# Estimating Regional Methane Emission Factors from Energy and Agricultural Sector Sources Using a Portable Measurement System: Case Study of the Denver–Julesburg Basin

**DOI:** 10.3390/s22197410

**Published:** 2022-09-29

**Authors:** Stuart N. Riddick, Fancy Cheptonui, Kexin Yuan, Mercy Mbua, Rachel Day, Timothy L. Vaughn, Aidan Duggan, Kristine E. Bennett, Daniel J. Zimmerle

**Affiliations:** 1The Energy Institute, Colorado State University, Fort Collins, CO 80524, USA; 2Cranfield Environment Centre, Cranfield University, Cranfield MK43 0AL, UK

**Keywords:** GHG inventory, emission factors, mobile survey, energy, agriculture

## Abstract

Methane (CH_4_), a powerful greenhouse gas (GHG), has been identified as a key target for emission reduction in the Paris agreement, but it is not currently clear where efforts should be focused to make the greatest impact. Currently, activity data and standard emission factors (EF) are used to generate GHG emission inventories. Many of the EFs are globally uniform and do not account for regional variability in industrial or agricultural practices and/or regulation. Regional EFs can be derived from top–down emissions measurements and used to make bespoke regional GHG emission inventories that account for geopolitical and social variability. However, most large-scale top–down approaches campaigns require significant investment. To address this, lower-cost driving surveys (DS) have been identified as a viable alternative to more established methods. DSs can take top–down measurements of many emission sources in a relatively short period of time, albeit with a higher uncertainty. To investigate the use of a portable measurement system, a 2260 km DS was conducted throughout the Denver–Julesburg Basin (DJB). The DJB covers an area of 8000 km^2^ north of Denver, CO and is densely populated with CH_4_ emission sources, including oil and gas (O and G) operations, agricultural operations (AGOs), lakes and reservoirs. During the DS, 157 individual CH_4_ emission sources were detected; 51%, 43% and 4% of sources were AGOs, O and G operations, and natural sources, respectively. Methane emissions from each source were quantified using downwind concentration and meteorological data and AGOs and O and G operations represented nearly all the CH_4_ emissions in the DJB, accounting for 54% and 37% of the total emission, respectively. Operations with similar emission sources were grouped together and average facility emission estimates were generated. For agricultural sources, emissions from feedlot cattle, dairy cows and sheep were estimated at 5, 31 and 1 g CH_4_ head^−1^ h^−1^, all of which agreed with published values taken from focused measurement campaigns. Similarly, for O and G average emissions for well pads, compressor stations and gas processing plants (0.5, 14 and 110 kg CH_4_ facility^−1^ h^−1^) were in reasonable agreement with emission estimates from intensive measurement campaigns. A comparison of our basin wide O and G emissions to measurements taken a decade ago show a decrease of a factor of three, which can feasibly be explained by changes to O and G regulation over the past 10 years, while emissions from AGOs have remained constant over the same time period. Our data suggest that DSs could be a low-cost alternative to traditional measurement campaigns and used to screen many emission sources within a region to derive representative regionally specific and time-sensitive EFs. The key benefit of the DS is that many regions can be screened and emission reduction targets identified where regional EFs are noticeably larger than the regional, national or global averages.

## 1. Introduction

Methane (CH_4_) is a powerful greenhouse gas with a greenhouse warming potential 84 times that of carbon dioxide over 20 years and is also partially responsible for the production and loss of tropospheric ozone. Since 1850, atmospheric CH_4_ mixing ratios have increased from 715 ppb to 1896 ppb in 2021 [1,2]. This increase in mixing ratio is largely attributed to increased anthropogenic emissions [3]. The ability to estimate the size and location of CH_4_ emissions is essential for all mitigation strategies and associated policies [4]. The majority of current national greenhouse gas (GHG) emission inventories are compiled using recommended emission factors (EFs) and estimates of activity levels [5,6,7]. Despite their widespread use, recent studies suggest that the use of emission factors may be insufficient to describe CH_4_ emissions from complex processes because many drivers of emissions are changing environmental conditions, such as temperature, or are the result of regulatory change that can vary between neighboring state or countries [8,9,10,11,12,13,14,15]. One solution to single EFs is to use regional EFs that are empirically generated and reflect the environmental and regulatory differences. However, generating regional EFs is challenging and requires quantifying emissions from enough (~5%) facilities within the region to make the EF representative.

Emission quantification methods include Gaussian-based plume approaches [8,12,16,17,18,19], backward Lagrangian stochastic (bLs) dispersion modeling [12,20,21,22], tracer flux methods [14,23,24,25], mass balance approaches [26,27,28], and remote sensing from aircraft [29,30,31] or satellites [32]. Of these approaches, emission estimates generated by the tracer flux method are reported to be the most accurate, ±20% [24], but the approach requires favorable winds (strong, but not too strong and blowing a direction accessible via roadway) and measurements can take a long time (~4 h per site measurement). Mass balance measurements with a CH_4_ analyzer mounted on an aircraft [27] or drone [26] can give relatively accurate results, ±50% [33,34], can be performed faster than tracer flux measurements (~1 h per site), but still an hour to measure each site. Remote sensing, either using aircraft or satellite, is becoming more popular as it can observe the emissions from sites in hundreds of km in day; however, detection limits are much higher than the other methods with 10+ kg CH_4_ h^−1^ for aircraft [30], 100+ kg CH_4_ h^−1^ for satellites [32], and therefore will not be able to quantify the majority of emission sources. Another major shortcoming of the tracer flux, mass balance, aircraft and satellite methods is that instrumentation is expensive and requires significant expertise to operate them and retrieve data. Therefore, these are relatively unrealistic options for everyone except those running well-funded and dedicated research facilities.

The Gaussian modelling approach has been used in ground-based, vehicle-mounted measurement campaigns to estimate CH_4_ emissions from individual O and G operations in the US and Canada [35,36,37,38]. Driving surveys (DS) are conducted by mounting a trace CH_4_ analyzer in a vehicle and calculating emissions using matching meteorology in an atmospheric dispersion model. This means that DSs are relatively inexpensive to equip and the measurement teams require little instrument expertise. The vehicle measuring CH_4_ concentrations travels at between 20 and 30 km h^−1^ downwind of a source and the highest measured concentration is assumed to be directly downwind of the point source [35,36,37]. The emission rate can then be calculated from the distance from the source, wind speed and an estimate of the atmospheric stability, the associated emission uncertainty has been reported at ±63% [39] and shown to decrease on repeat measurements [36,40]. The shortcoming of the Gaussian approach is that it can only be used to estimate emissions from point sources and the reported uncertainties are based on repeat measurements and seem unrealistically small for a single observation. Backwards Lagrangian stochastic methods can be parameterized to estimate emissions from area sources and use much the same input as the Gaussian approach; however, they have never been used as part of a DS.

This study aims to investigate whether DSs, using Gaussian and bLs approaches to generate emission estimates, can be used to sample enough sources to generate representative regional emission factors. Driving surveys present a relatively inexpensive (labor and equipment) method to identify individual sources and could provide insight into the apportionment of emission source, but come with inherent, possibly large, uncertainties in calculated emission. To investigate this, our objectives were to (1) conduct DSs in the DJB; (2) estimate the CH_4_ emissions from individual sources detected; and (3) compare emission factors and regional emissions derived by the DS to other estimates calculated from focused campaigns to investigate the strengths/weaknesses of the DS.

## 2. Materials and Methods

### 2.1. Driving Survey

The DS was conducted in the Denver–Julesburg Basin (DJB), CO on 10 days between 4 July and 18 July 2021. The DJB is a geological structural basin that contains oil and gas deposits in an area covering 8000 km^2^ and is located between Denver and Pierce, CO. Oil was first discovered there in 1862 [41], and by 2014, there were 24,000 oil and gas wells in the DJB producing over 90% of Colorado’s oil [42]. In 2021, as per the Colorado Department of Public Health and Environment (CDPHE) and Colorado Oil and Gas Conservation Commission (COGCC) database, there were 72,930 wellheads (COGCC, unpublished data) in operating in Weld County on 8176 individual well pads (CDPHE, unpublished data). In 2014, oil and gas operations in the DJB were identified as the most significant source of CH_4_ emissions (75% of total emission), as identified by the CH_4_ to ethane ratio in air samples collected in the region [43]. Other CH_4_ sources of emissions within the DJB include 74 registered agricultural operations (AGOs), 32 waste-water treatment facilities and 3 landfills. The AGOs comprise 409,550 cattle, 3080 broiler chickens, 48,000 laying chickens, 2880 pullet chickens, 184,643 dairy cows, 336 horses, and 27,000 sheep (unpublished data, CDPHE, 2021). Additionally, it is unclear if smaller AGOs are included in these counts, as smaller operations do not necessarily report to state or federal programs.

The 2260 km route (Figure 1) was driven at between 20 and 30 km h^−1^. Methane concentrations were measured by either an ABB LGR-ICOS GLA132 ultra-portable or ABB GLA131-GGA micro-portable greenhouse gas analyzer (MGGA). Both analyzers are laser absorption spectrometers that measure CH_4_ mole fractions in air [44] and report CH_4_ mole fractions every second, with a stated precision of <2 ppb (1σ at 1 Hz) over an operating range of 0.1 to 100 ppm. The inlet line was attached to the car to avoid contamination from the exhaust and protected from water incursion. The air intake was filtered using a 2 µm filter. Meteorological data were collected using Airmar 150WX ultrasonic weather station (Airmar, Milford, CT, USA). The weather station was on the end of a 2 m mast attached to the back of the car and above the dead air zone. Meteorological data were recorded at 1 s or 5 s intervals and included wind speed (*u*, m s^−1^), wind direction (*WD*, ° to North), air temperature at 2 m (*T_a_*, K), relative humidity (*RH*, %) and air pressure (*P*, Pa).

### 2.2. Source Detection

Emission sources were identified from the mixing ratio data collected by the CH_4_ analyzers. Due to the dynamic nature of the background concentration, an emission source was identified using a peak-finding algorithm [45]. To avoid signal-to-noise issues caused by the changing background concentration, the algorithm first smooths the first derivative of the signal, before identifying where the derivative crosses the *x*-axis at a slope greater than 10 ppb s^−1^. Each peak identified by the software was investigated by eye and identified as either a point source or an area source. In the case of a point source, the time and maximum concentration were recorded (Appendix A). For area sources, the time, location, distance observing the enhancement and the average enhancement were recorded (Appendix A).

### 2.3. Quantifying Emissions

Methods for quantifying point source and area source emissions differ. Point source emissions were quantified using a Gaussian plume approach [46], where the emission (Q, g s^−1^) was calculated from the location, peak measured enhancement, wind speed, and atmospheric stability data. This method has been used to estimate emissions from oil and gas infrastructure in recent studies [12,36]. Area emissions were calculated using a bLs method, using source-detection distance, average measured CH_4_ enhancement, dimension of the emission source (estimated from satellite images), wind speed, and atmospheric stability data [22,47]. The WindTrax model (www.thunderbeachscientific.com accessed on 1 February 2022) has been used to estimate emissions from oil and gas infrastructure [11] as well as natural [48,49], waste [20], and agricultural sources [50,51].

#### 2.3.1. Point Sources

The Gaussian Plume (GP) model calculates the mole fraction of a gas as a function of distance downwind from a point source [46,52]. When CH_4_ is emitted from a point source, it enters the air flow and disperses vertically and laterally with time, forming a cone. The CH_4_ enhancement above background (*Χ*, μg m^−3^) *x* meters downwind, *y* meters from the center of the plume, and *z* meters above ground level can be calculated from the emission rate (*Q*, g s^−1^), the height of the source (*h_s_*, m), the height of the boundary layer (*h*, m) and the stability of the air [46]:(1)$$X\left(x,y,z\right)=\frac{Q}{2\pi u\sigma_{y}\sigma_{z}}e^{\frac{-y^{2}}{2\sigma_{y}{}^{2}}}\left(e^{\frac{-{\left(z-h_{s}\right)}^{2}}{2\sigma_{z}{}^{2}}}+e^{\frac{-{\left(z+h_{s}\right)}^{2}}{2\sigma_{z}{}^{2}}}+e^{\frac{-{\left(z-2h+h_{s}\right)}^{2}}{2\sigma_{z}{}^{2}}}+e^{\frac{-{\left(z+2h-h_{s}\right)}^{2}}{2\sigma_{z}{}^{2}}}+e^{\frac{-{\left(z-2h-h_{s}\right)}^{2}}{2\sigma_{z}{}^{2}}}\right)$$

In the case that the maximum enhancement is used, it is assumed that measurement is at the center of the plume laterally and *y* = 0 m. The standard deviation of the lateral (*σ_y_*, m) and vertical (*σ_z_*, m) mixing ratio distributions are calculated from the Pasquill–Gifford stability class (PGSC) of the air [46,53,54]. The following assumptions were made: (1) the source is emitting at a constant rate; (2) mass of CH_4_ is conserved when reflected at the surface of the ground or the top of the boundary layer; (3) wind speed and vertical eddy diffusivity are constant with time; (4) there is uniform vertical mixing; and (5) the terrain is relatively flat between source and detector.

##### Model Input

For input to the GP equation above, some generalized assumptions were used. It was assumed that any single peak was a point source, and emission heights for sources on well pads, compressor stations, and gas plants were 1.5, 8 and 10 m above the ground, respectively. PGSC were assigned as a function of wind speed and solar irradiance [52,54] and a full description of how PGSC is assigned is presented in Appendix A. Irradiance was based on observation, measurements at Colorado State University’s meteorological site at Christman Field and always defined as strong, as is typical during the day in the Colorado summer. PGSC were then assigned as A for wind speeds less than 2 m s^−1^, B for wind speeds between 2 and 5 m s^−1^ and C for wind speeds greater than 5 m s^−1^. An example emission calculation for a point source is presented in Appendix A.

##### Uncertainties

To evaluate the uncertainty of our experimental setup, a controlled release uncertainty analysis was performed at Colorado State University’s Methane Emissions Technology Evaluation Center (METEC) facility in Fort Collins, CO, USA. Compressed natural gas, with methane compositions ranging from 85 to 95%vol supplied from two 145 L cylinders and flow rates controlled using a pressure regulator and precision orifices, was released from the end of Teflon tubing 1.5 m above the ground at the center of the site at 0.25, 0.86 and 2.6 kg CH_4_ h^−1^. At METEC the methane content of the natural gas in each release is measured by gas chromatography and accounted for in the known emission rate. The MGGA was mounted in a vehicle following the method described in Section 2.1 and driven round the METEC site ten times for each emission rate at the maximum speed allowed (16 km h^−1^). Emission rates were then calculated following Section 2.2 and Section 2.3, and the percentage uncertainty was calculated from the known and measured/calculated emission rates. The measurement uncertainty is presented as the 95% confidence interval of the 10 repeat measurements.

#### 2.3.2. Area Sources

Backward Lagrangian stochastic (bLs) models, such as WindTrax, model the path of a gas in the atmosphere as it moves away from an area source [22,47]. The CH_4_ emission rate from an area is calculated by modeling how the path of thousands of CH_4_ particles are affected by horizontal and vertical aerodynamic forces in the boundary layer. A bLs runs a simulation to determine the ratio of expected measured concentration to emission rate for given meteorological and micrometeorological conditions and scales according to the actual measured concentration to generate an emission rate [22,47].

WindTrax requires data on source size, measured CH_4_ enhancement, wind speed and atmospheric stability. A practicable terrain between the source and analyzer is a fetch with roughness length less than 15 cm [55,56] and a maximum downwind distance of 1 km [47,50]. In this study, WindTrax atmospheric dispersion model version 2.0.8.8 was used in the inverse mode to infer the CH_4_ emissions from area sources. Each of the averaged enhancements were used as input data to back-calculate the CH_4_ emission using 50,000 particle projections. Data used as input to WindTrax were wind speed (*u*, m s^−1^), wind direction (*WD*, °), temperature (*T*, °C), average CH_4_ enhancement at 1 m (*X*, μg m^−3^), and the Pasquill–Gifford atmospheric stability class. The roughness length was estimated from observation using ‘‘tall grass” (*z*_0_ = 10 cm) in the surface data sub-menu of WindTrax. The Pasquill–Gifford atmospheric stability class (A–F) was assigned using the same method as described above [52].

It was assumed the area emission sources are on the ground (*z* = 0 m). The area of the source and the upwind distance from the road were identified from Google satellite imagery. The roughness length was taken as long grass (*z*_0_ of 10 cm). As with the GP approach, the uncertainty associated with an emission estimate generated by this approach has never been measured. Controlled release studies estimate the emission uncertainty using the bLs method for a stationary instrument at ±12% [57], but it is reasonable to assume that an emission calculated from measurements in a moving vehicle will be considerably larger and consistent with the uncertainty estimated for point sources. An example emission calculation for an area source is presented in Appendix A.

## 3. Results

### 3.1. Uncertainty Analysis

The uncertainty analysis experiment was conducted on 1 August 2022 between 9 and 11:30 a.m. Average maximum mixing ratios for each 10 measurements were 2.25, 2.89 and 6.13 ppm for the 0.25, 0.86 and 2.6 kg CH_4_ h^−1^ releases, respectively (Figure 2). Using the methods described in Section 2.2 and Section 2.3, average emissions were calculated at 0.24, 0.89 and 2.65 kg CH_4_ h^−1^. Using 95% confidence intervals, the uncertainty is estimated at +40.7%/−42.3%, +60.0%/−56.8% and +43.0%/−42.3% for the 0.25, 0.86 and 2.6 kg CH_4_ h^−1^ releases, respectively. Accordingly, uncertainty bounds of ±60% will be used to indicate the minimum accuracy of a single emission measurement made in this study. Environmental factors, such as atmospheric stability, varying roughness lengths, and aerodynamic obstructions, can confound the airflow from the source to the detector and result in much larger uncertainty in emission quantification. Here, we present this uncertainty estimate as the variability in emission quantification from the experimental setup.

### 3.2. Sources Detected

During the DS, we estimate 534 sources could have been observed (82 from AGOs, 6 natural and 444 from O and G); of those, 157 individual CH_4_ emission sources were detected (Table 1). In total, 52% of the sources were from agricultural sources (feedlot cattle, dairy farms, sheep or irrigation ponds), 44% from O and G operations (compressor stations, gas plants, pipelines and well pads), and 4% from natural sources (e.g., lakes and wetlands). Of the AGOs, 51% of plumes were from CAFOs, 29% came from dairy operations, 17% from irrigation ponds, and 2% from sheep operations. For O and G operations, emissions from well pads made up 78% of the plumes detected during the two DSs, gas plants made up 14%, while compressors and pipelines contributed 3 and 4%, respectively (Table 1). Pipeline emissions were identified using satellite images, where emissions were observed from above ground infrastructure or metering stations.

### 3.3. Emissions

The emissions from the sources show a typical long-tail distribution observed in many other CH_4_ emissions measurement studies, i.e., many smaller emitters and relatively small number of large emitters [58,59,60]. The lowest calculated emission observed was 20 g CH_4_ h^−1^, where an enhancement of 236 ppbv was detected 60 m away from a well pad comprising 4 wells heads, 8 separators, 4 water tanks, 18 condensate tanks and 5 combustors. We suggest this is the lower detection limit of this method, i.e., sources emitting less than 20 g CH_4_ h^−1^, cannot be detected using a driving method. The largest calculated emission observed was 327 kg CH_4_ h^−1^ from a gas plant, where an enhancement of 106 ppbv was detected 100 m downwind of a gas processing plant. Another limitation of these observations is this method does not provide any information on the duration of the emission and cannot be used to distinguish between short-duration events, e.g., a blowdown or something more persistent such as an uncontrolled emission event.

The average CH_4_ emission from AGOs was 30, 77, 2 and 25 kg CH_4_ site h^−1^ for cattle feedlots, dairy farms, irrigation ponds and sheep enclosures, respectively. The average emissions from O and G operation are estimated at 14, 110, 20 and 0.3 kg CH_4_ site h^−1^ for compressor stations, gas processing plants, pipeline emissions and well pads, respectively. Of the total CH_4_ observed during the DS, AGOs and O and G represented nearly all of the CH_4_ emissions in the DJB, accounting for 58% and 32% of the total emission. Natural sources of emission, reservoirs and lakes, account for 10% of the CH_4_ emission observed.

### 3.4. Representative Published Emission

Methane emissions from feedlot cattle, dairy cows and sheep were estimated at 9.4, 39.3 and 0.9 g head^−1^ h^−1^ at agriculture operations in the DJB in 2014 [61]. Methane emissions for open water are estimated at 2 (1–66) mg CH_4_ m^2^ h^−1^ [62]; these were measured at two shallow lakes similar to those found in the DJB with the range in emissions taken from the variability observed during the summer months. Average emissions of 9 and 237 kg CH_4_ h^−1^ were from compressor stations and gas processing plants in the DJB [63,64] using a tracer flux method, which has an associated uncertainty of ±20% [24]. The US EPA’s Other Test Method (OTM) 33A was used to quantify short-term emission rates from 210 oil and gas production pads during eight two-week field studies in Texas, Colorado, and Wyoming from 2010 to 2013, emission rates from O and G production pads in the DJB were estimated at 0.14 g s^−1^ [65].

In 2012, Pétron et al. (2014) used aircraft-based CH_4_ observations in a mass–balance flux calculation to estimate total CH_4_ emissions in the DJB at 26.0 t CH_4_ h^−1^. Using inventory data, they estimated total CH_4_ emissions from four non-O and G gas sources (AGO, natural and waste) at 6.7 t CH_4_ h^−1^. The difference between the top–down estimate and the bottom-up non-O and G emission estimate was then attributed to O and G sources in the DJB with a total O and G CH_4_ emission estimate of 19.3 t CH_4_ h^−1^.

### 3.5. Driving Survey EFs

The Colorado Department for Public Health and Environment (CDPHE) data show there are 74 cattle operations in the DJB with a total of 410,000 cattle. The DS estimates the average emission from 42 cattle facilities at 679 kg CH_4_ facility^−1^ h^−1^ resulting in an EF of 5.3 g CH_4_ animal^−1^ h^−1^. For dairy cows, there are 74 operations with a total of 185,000 mature dairy cows and measurement estimates 1540 kg CH_4_ facility^−1^ h^−1^ resulting in an EF of 30.8 g CH_4_ animal^−1^ h^−1^. The three sheep operations in the DJB have a total of 27,000 sheep resulting in an average emission of 0.9 g CH_4_ animal^−1^ h^−1^. The eight water bodies measured in the driving campaign covered 6,500,000 m^2^, measurement estimates total emission 100 kg CH_4_ h^−1^ from this resulting in and EF of 0.015 g CH_4_ m^−2^ h^−1^. The average emissions from compressor stations, gas plants and well pads were measured at 14, 110 and 0.28 kg CH_4_ facility^−1^ h^−1^, respectively (Table 2).

## 4. Discussion

This study investigated using a DS to generate regionally representative EFs for CH_4_ emissions sources in agriculture and energy sectors. A 2260 km DS was conducted over 10 days throughout the Denver–Julesburg Basin, which is densely populated with O and G operations and AGOs, to detect individual emission sources and use dispersion modelling to infer the emission rate of each. Controlled release experiments suggest the minimum uncertainty, based on 95% confidence intervals, in source emission quantification using a single measurement is ±60%. It should be noted that this is in ideal conditions where the fetch between the source and detector is flat, and the speed was limited to 16 km h^−1^ due to site safety restrictions. Regardless, repeat experiment show negligible bias in the quantification method.

During the survey, 157 individual CH_4_ emission sources were detected, including 82 from AGOs, 69 from O and G operations, and 6 from natural sources. Of the total CH_4_ observed during the DS, AGOs and O and G represented nearly all of the CH_4_ emissions in the DJB, accounting for 58% and 32% of the total emission. Natural sources of emission, reservoirs and lakes, account for 10% of the CH_4_ emission observed. For AGOs, feedlot cattle, dairy cows and sheep EFs are estimated at 5, 31 and 1 g CH_4_ head^−1^ h^−1^, respectively. EFs and uncertainty ranges for compressor stations, gas processing plants and well pads are estimated at 14, 110 and 0.28 kg CH_4_ facility^−1^ h^−1^, respectively. One key limitation of the DS is the inability to differentiate between duration of events, where short-duration events, e.g., a blowdown, may confound the quantification of a more persistent event, such as an uncontrolled emission event. Here, the relatively low-cost of the DS may be further used to repeat the route to note large emissions events that are both persistent and unexpected.

Generally, the average emission observed by intense field studies focused on a single source in the DJ basin [14,61,66,67] fall within the uncertainty bounds of the DS EFs. Confidence is not strong in the gas plant’s EF (Table 1), as a potentially incorrect assumption was made about the height of the emission source and the DS only provides a short duration measurement that may not account for maintenance activities or short duration emission events. Gas plants aside, the data calculated from the DS’s observations could provide a useful “first look” at regional emissions and help identify targets for CH_4_ mitigation or sources of interest for further and more intensive field studies. As the DS simply requires a trace gas analyzer and an instrument to measure the wind fields, it is a relatively simple data collection exercise when compared to tracer release or mass balance measurements.

The disparity in the gas processing plant emissions reflect the relative difficulty in generating an EF for a source where emissions are defined by operational activity and regulation, instead of biogeochemical responses such as agricultural emission. Emissions based on activity are likely to vary greatly throughout the day depending on oil production throughput or the presence of process upset conditions, with larger emissions being the result of venting (flashing and off-gassing) from liquid storage tanks, leaks/venting from compression equipment and gas pneumatics, or flaring from process equipment [66]. Therefore, “snapshot” emission estimate generated by a DS may not be representative of the actual emissions; the low-cost of the driving survey mean that repeat measurements could be made to produce better estimates, and reduce overall uncertainty. Alternatively, representative EF may be reconciled through a targeted measurement approach, where fence-line sensors could be deployed and the temporal uncertainty smoothed when the average emission is calculated from a longer time-series [17,68].

Unlike processing plants, well pads in the DJB are plentiful, and a representative EF could be generated by measuring “snapshot” emissions at many sites. It is estimated that this DS observed emissions from 5% of the well pads (432 of a total 8176 well pads), with measurable emissions coming from only 12% of the wells observed. Brantley et al. (2014) reported mean emission rate of 504 g CH_4_ h^−1^ from DJB production pads and we estimate emissions at 280 g CH_4_ h^−1^. Our low estimate may, in part, reflect the relatively tough emission regulation in the DJB and company initiatives to reduce emissions including return lines or flaring during condensate unloading, increased use of three-stage separation on well pads and/or removal of atmospheric tanks, and increased requirements for use of leak detection and repair surveys. Results may also reflect the economy in 2021, where a number of wells in the DJB were temporarily shut-in and production halted, this means that any emissions from leaking equipment would not have been observed by this study. In 2014, O and G operations were identified as the largest CH_4_ emission sector in the DJB emitting 19.3 Mg CH_4_ h^−1^ while other sources contributed 6.7 Mg CH_4_ h^−1^ [43]. The results of our DS suggest that, in 2021, this has changed, with AGOs emitting a similar amount to 2014 (7.9 Mg CH_4_ h^−1^) while emissions from O and G of 6.4 Mg CH_4_ h^−1^, are reduced to one third. This again could be explained by high levels of regulation of the O and G industry implemented in the DJB over the last 10 years, the shutting-in of operations which have reduced basin-wide CH_4_ emissions or a shift in production operations to larger, horizontal drilling pads with more separation stages and better overall control systems instead of older, smaller well pads. Regardless of the specific reasons, this highlights the effectiveness of DSs on capturing a time-dependent EF.

Using CDPHE activity data and EFs derived by the DS (Table 2), we can extrapolate to an estimated annual DJB CH_4_ emission of 125 (63–250) Gg CH_4_ yr^−1^ with the main emission sector identified as agriculture and dairy farming in particular. Our estimate is lower than the 210 Gg CH_4_ emission per year estimated by extrapolating the findings of Pétron et al. (2014), which may reflect our underestimation in emissions from gas plants and compressor station. Our estimate also omits to report emissions from the waste sector as no emissions from landfill or wastewater treatment facilities were observed during the driving study. Despite this, the general agreement between our basin-wide emission estimate and previous estimates suggest relatively short DSs could be a low-cost alternative to traditional measurements campaign and used to screen many emission sources within a region to derive representative EFs from individual sources. The key benefit of adopting this approach is that many sources within a region can be screened and mitigation efforts targeted on specific regional activities where emissions are noticeably larger than the regional, national or global averages. However, care should be taken with the observations as the relatively high uncertainty may mean that observation is not particularly useful for detecting subtle differences in emissions, such as farming management strategies where variability could remain in the noise of the measurement. Therefore, DSs may have more utility in detecting variability in emission caused by a response to regulation and a useful tool in observing and monitoring regulation and mitigation strategies.

## 5. Conclusions

Direct measurements made during this study finds that a portable CH_4_ mixing ratio/meteorological measurement system can be used to quantify the emissions from over 500 individual sources from the agricultural, energy and nature emission sectors over a 10-day period with an accuracy of ±60%. Within the DJB, it was found that AGOs and O and G operations represented nearly all the observed CH_4_ emissions, accounting for 54% and 37% of the total emission, respectively. Average facility emissions from cattle feedlots and dairy cow operations (30 and 77 kg CH_4_ facility h^−1^, respectively) were larger than from O and G well pads and compressor stations (0.3 and 14 kg CH_4_ facility h^−1^, respectively) and gas processing plants (0.5, 14 and 110 kg CH_4_ facility^−1^ h^−1^) but smaller than gas processing plants (110 kg CH_4_ facility h^−1^). All average emissions were in reasonable agreement with emission estimates from intensive measurement campaigns. A comparison of our basin wide O and G emissions to measurements taken a decade ago show a decrease of a factor of three, which can feasibly be explained by changes to O and G regulation over the past 10 years, while emissions from AGOs have remained constant over the same time period. Our data suggest that DSs could be a low-cost alternative to traditional measurement campaigns and used to screen many emission sources within a region to derive representative regionally specific and time-sensitive EFs. The key benefit of the DS is that many regions can be screened and emission reduction targets identified where regional EFs are noticeably larger than the regional, national or global averages.

## Figures and Tables

**Figure 1 sensors-22-07410-f001:**
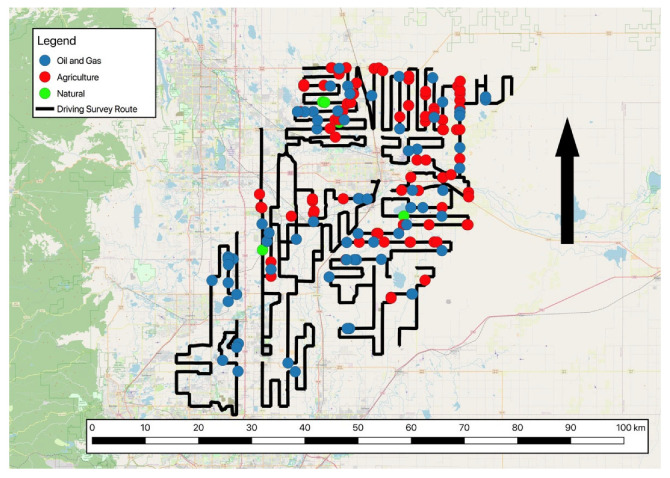
Driving survey route in the Denver–Julesburg basin and locations of natural, oil and gas and agricultural sources detected. The arrow shows North.

**Figure 2 sensors-22-07410-f002:**
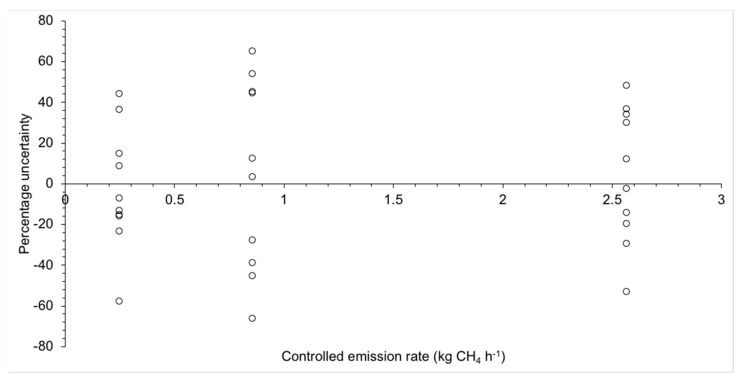
Percentage uncertainty, calculated from known and measured emission rates, of controlled release uncertainty analysis performed at Colorado State University’s Methane Emissions Technology Evaluation Center (METEC) facility in Fort Collins, CO, USA. Compressed natural gas was released at 1.5 m above the ground at the center of the site at 0.25, 0.86 and 2.6 kg CH_4_ h^−1^ and mixing ratios measured inside a vehicle moving around the site at 16 km h^−1^.

**Table 1 sensors-22-07410-t001:** The number and total emission estimates from sectors in the DJ Basin detected during weekend/weekday surveys. Sub-sources for sectors or presented in italics and uncertainties in average emission in brackets.

Sector	Number of Sources Sampled	Number of Plumes Detected	Total Emission (kg h^−1^)	Average Emission (kg Facility^−1^ h^−1^)
Agriculture	82	82	2326	
*Feedlot cattle*	*42*	*42*	*679*	*29.5 (47.2, 11.8)*
*Dairy*	*20*	*20*	*1540*	*77.0 (123.2, 30.8)*
*Livestock*	*23*	*23*	*35*	*1.5 (2.4, 0.6)*
*Irrigation ponds*	*14*	*14*	*21*	*1.5 (2.4, 0.6)*
*Sheep*	*2*	*2*	*50*	*25.1 (40.2, 10.0)*
Natural	6	6	362	
*Reservoirs*	*5*	*5*	*232*	*46.5 (74.4, 18.6)*
*Lakes*	*1*	*1*	*130*	*129.7 (207.5, 51.9)*
Oil and Gas	444 *	69	1307	
*Compressor Station*	*2*	*2*	*28*	*14.0 (22.4, 5.6)*
*Gas Plant*	*10*	*10*	*1097*	*109.7 (175.5, 43.9)*
*Pipelines*		*3*	*61*	*20.2 (32.3, 8.1)*
*Well pads*	*432*	*54*	*121*	*0.28 (0.45, 0.11)*
Total		157	3995	

* Estimate based on GIS of all wells (active and shut in) in the CDPHE and COGCC databases.

**Table 2 sensors-22-07410-t002:** Comparison of emission factors (EFs) for sources in the Denver–Julesburg basin as calculated from individual measurement campaigns (Published EF) and those generated by this study (Observed EF). N.B. “Comp” denotes compressor stations.

Source	EF Unit	Published EF	Driving Survey EF	Activity (Count)	Emission (Mg h^−1^)	Emission (Gg y^−1^)
Cattle	g CH_4_ head^−1^ h^−1^	9.4	5.3 (2.1–8.5)	409,550	2.2	19
Dairy	g CH_4_ head^−1^ h^−1^	39.3	31 (12–50)	184,463	5.7	50
Sheep	g CH_4_ head^−1^ h^−1^	0.9	0.9 (0.4–1.4)	27,000	0.02	0.2
Lakes	mg CH_4_ m^−2^ h^−1^	2	15 (6–24)			
Comp	kg CH_4_ facility^−1^ h^−1^	9	14 (6–22)	64	0.9	8
Gas plant	kg CH_4_ facility^−1^ h^−1^	237	110 (44–176)	29	3.2	28
Well pad	g CH_4_ facility^−1^ h^−1^	504	280 (112–448)	8176	2.3	20
Total						125 (63–250)

## Data Availability

Data can be accessed by contacting the corresponding author at stuart.riddick@colostate.edu.

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
