# Peer review of "Estimating Regional Methane Emission Factors from Energy and Agricultural Sector Sources Using a Portable Measurement System: Case Study of the Denver–Julesburg Basin"

_sensors, 2022, doi:10.3390/s22197410_

Round 1

Reviewer 1 Report

This paper summarizes the findings of a repeat drive around survey in Northern Colorado. Emissions sectors are broken down into three categories including agriculture, oil and gas and natural sources. The results of this work are generally consistent with other work in this region. The results are rather routine, but the analysis is robust. Overall I recommend publication following minor changes. In particular, I draw attention to the use and interpretation of some citations.

Specific Comments

L 66 There are actual citations for aircraft mass balance uncertainty that could be used rather than a personal communication. I don’t believe Scientific Aviation has ever published a paper with a definitive uncertainty analysis for their methodology, though they may have some examples. Relevant citation to aircraft mass-balance uncertainty include Cambaliza et al. 2014 (doi:10.5194/acp-14-9029-2014) and Heimburger et al.  2017 (doi.org/10.1525/elementa.134). I would also prefer consistency between the way uncertainty is presented for all techniques. Some techniques have one number, others have a range.

L 78 Regarding the driving surveys as a novel strategy, Albertson et al. 2016 also discussed using driving survey to efficiently sample emissions (doi.org/10.1021/acs.est.5b05059).

L 86 How did you read the cited literature and arrive at 33% uncertainty from these approaches? MacKay et al reports 63% uncertainty on emission calculation and Caulton et al reports much higher uncertainties that are dependent on the number of repeat passes. Also see Raznjevic et al 2022 who reported 40% uncertainty can be achieved with 15 repeat transects https://doi.org/10.5194/amt-15-3611-2022

L 151 You give examples of Gaussian derived emissions that include a mobile ground technique, mobile boat technique and stationary ground technique. The stationary and mobile techniques are sufficiently different that I find the grouping of these citations unclear. Is this supposed to suggest that you use some aspects of the OTM method? That doesn’t seem to be the case to me.

L 191 “The most relevant study to our measurement is Edie at al. (2020)…” No, because this is the uncertainty for a different technique. Just because they both use the Gaussian model does not make these approaches equivalent. By its nature, the stationary OTM technique requires longer data collection. The fact that you can calculate an emission with one pass with a mobile technique makes this quite different. It seems like the only thing is relevant is that they did a controlled release. I would point out that Rella et al. 2015 (doi.org/10.1021/acs.est.5b00099), Yacovitch et al. 2015 (doi.org/10.1021/es506352j), and Caulton et al. 2018 (doi.org/10.5194/acp-18-15145-2018) also did controlled releases.

L 244 The use of the controlled release as the only way to calculate uncertainty seems overly simplistic. The exact location and height of the control release may be known (it’s unclear if these were blind releases). I would expect that the uncertainty in these scenarios would represent the minimum uncertainty possible. Also, only three releases were performed (presumably over 1 day?). As other studies have reported, meteorological conditions can affect plume variability and thus the uncertainty that would be calculated with this method. I would like to see some additional explanation of these other factors and the limitations of this approach.

Table 1. Can you say anything about the prevalence of sources vs number of sites sampled? For example, it seems like only ~12% of well pads had a measurable signal. That’s quite low. Is that meaningful, or are some of those sites potentially too far to be measured anyway? Or for example, you detected 100% of the other animal types, but only about 50% of feedlot cattle, even though the feedlot cattle had high average emissions. Why might that be?

Reviewer 2 Report

This study estimates regional methane emission factors using a portable measurement system, which at the time is one of the low-cost and state-of-the-art technologies. The study measures DJB one of the important areas with many methane sources in US and shows the benefits of portable measurement systems. This study is overall well designed and written, and I only have some minor concerns of it.

1.       It seems that most authors are native English speakers, but please be careful to examine some small grammar errors, for examples:

(1)    Line 10: Methane (CH4) a powerful greenhouse gas (GHG) it has been

(2)    Line 27-28

(3)    Line 68: 100s km in day (maybe just hundreds.)

(4)    Line 72, are>- is

2.       Figure 1, could you show the detected sources on the map and categorize them?

3.       It will also help the readers if there are some pictures/demonstrations/examples of how you drive the DS across point or area sources

4.       It you can provide some codes on the method of computing emission rates from both point and area sources, that will be even better for people new to this area to apply or adopt your methods.

5.       More details are probably needed for the PGCS.

6.       Line 312 what is CDPHE?

7.       Line 300, what is OTM 33A?
